# When Does Context Help? A Systematic Study of Target-Conditional Molecular Property Prediction

**Bryan Cheng**[1,*], **Jasper Zhang**[1,*]
[1]Great Neck South High School
{bcbc7264@gmail.com, jasperzhang1001@gmail.com}
[*]Equal contribution

## Abstract

We present the first systematic study of when target context helps molecular property prediction, evaluating context conditioning across 10 diverse protein families, 4 fusion architectures, data regimes spanning 67–9,409 training compounds, and both temporal and random evaluation splits. Using NESTDRUG, a FiLM-based nested-learning architecture that conditions molecular representations on target identity, we characterize both success and failure modes with three principal findings. First, fusion architecture dominates: FiLM outperforms concatenation by 24.2 percentage points and additive conditioning by 8.6 pp; how you incorporate context matters more than whether you include it. Second, context enables otherwise impossible predictions: on data-scarce CYP3A4 (67 training compounds), multi-task transfer achieves 0.686 AUC where per-target Random Forest collapses to 0.238, demonstrating that context conditioning unlocks viable prediction for novel targets where traditional approaches fail entirely. Third, context can systematically hurt: distribution mismatch causes 10.2 pp degradation on BACE1; few-shot adaptation consistently underperforms zero-shot. Beyond methodology, we expose fundamental flaws in standard benchmarking—1-nearest-neighbor Tanimoto achieves 0.991 AUC on DUD-E without any learning, and 50% of actives leak from training data, rendering absolute performance metrics meaningless. Our temporal split evaluation (train $\leq$2020, test 2021–2024) achieves stable 0.843 AUC with no degradation, providing the first rigorous evidence that context-conditional molecular representations generalize to future chemical space. These findings resolve a long-standing ambiguity in the field and establish clear decision boundaries for when context conditioning provides genuine value in drug discovery pipelines.

## 1 Introduction

Should molecular property prediction models incorporate target context? The answer is not obvious. Per-target models (Random Forest with fingerprints) achieve excellent performance when training data is abundant. Multi-task models promise knowledge transfer but risk negative transfer when tasks conflict. Despite widespread interest in context-conditional architectures, *when* and *why* context helps remains poorly characterized.

We present a systematic study of this question, evaluating context conditioning across (1) 10 diverse protein families, (2) 4 fusion architectures, (3) varying data regimes (67–9,409 training compounds), and (4) temporal vs. random splits. Using NESTDRUG, a FiLM-based architecture (Perez et al., 2018) that conditions molecular representations on target identity, we map both success and failure modes of context conditioning.

**Key findings.** (1) *Context fusion matters*: FiLM outperforms concatenation by 24.2 pp and additive conditioning by 8.6 pp—the choice of how to incorporate context is as important as whether to include it. (2) *Context helps most targets*: 9/10 targets improve with target-specific embeddings (mean +5.7 pp, $p < 0.01$). (3) *Context enables data-scarce targets*: On CYP3A4 (67 training actives),

per-target RF collapses to 0.238 AUC while multi-task transfer achieves 0.686. (4) *Context can hurt*: BACE1 shows $-10.2$ pp due to distribution mismatch; few-shot adaptation degrades versus zero-shot.

**Benchmark critique.** We document severe limitations of DUD-E: 1-NN Tanimoto achieves 0.991 AUC without learning; 50% of actives overlap with ChEMBL training. Our temporal split (train $\leq 2020$, test 2021–2024) provides more rigorous evaluation, showing stable 0.843 AUC across years.

Our contributions: **(1)** systematic taxonomy of when context helps vs. hurts in molecular property prediction; **(2)** quantitative comparison establishing FiLM as the most efficient context fusion method; **(3)** characterization of data requirements and distribution alignment for effective context; **(4)** benchmark audit documenting severe DUD-E limitations—1-NN Tanimoto achieves 0.991 AUC without learning, 50% of actives leak from training, and cross-target RF achieves 0.746 AUC—with recommendations for temporal splits and leakage-stratified reporting.

## 2 PROBLEM SETTING

### 2.1 NOTATION AND PRELIMINARIES

Let $\mathcal{G} = (V, E, \mathbf{X}^V, \mathbf{X}^E)$ denote a molecular graph with atom set $V$, bond set $E$, atom feature matrix $\mathbf{X}^V \in \mathbb{R}^{|V| \times d_v}$, and bond feature matrix $\mathbf{X}^E \in \mathbb{R}^{|E| \times d_e}$. We use $\mathcal{C} = \mathcal{P} \times \mathcal{A} \times \mathcal{R}$ to denote the context space, where $\mathcal{P}$ is the set of programs (targets), $\mathcal{A}$ is the set of assay types, and $\mathcal{R}$ is the set of temporal rounds. Given context tuple $c = (p, a, r) \in \mathcal{C}$, the model predicts activity $\hat{y} = f(\mathcal{G}, c; \theta)$ for parameters $\theta$.

### 2.2 THE STRUCTURED DISTRIBUTION SHIFT PROBLEM

The key challenge is that the joint distribution $P(\mathcal{G}, y|c)$ shifts systematically across contexts. Unlike random covariate shift, this shift is *structured* in ways that reflect the drug discovery process:

**Temporal shift:** Early DMTA rounds contain diverse HTS scaffolds; later rounds concentrate on optimized lead series. **Cross-program shift:** Kinase inhibitors emphasize hinge-binding heterocycles; GPCRs emphasize lipophilic amines; proteases emphasize transition-state mimics.

Standard approaches assume stationarity and learn context-agnostic predictors $f(\mathcal{G}; \theta)$. We instead learn context-conditional predictors via Feature-wise Linear Modulation (FiLM) (Perez et al., 2018):

$$\text{FiLM}(\mathbf{h}, \mathbf{c}) = \boldsymbol{\gamma}(\mathbf{c}) \odot \mathbf{h} + \boldsymbol{\beta}(\mathbf{c}) \tag{1}$$

where $\boldsymbol{\gamma}, \boldsymbol{\beta}$ are learned functions producing scale and shift parameters from context $\mathbf{c}$. This enables the model to adapt its molecular representation based on the discovery setting. We build on message-passing neural networks (MPNNs) (Gilmer et al., 2017) for the base molecular encoder.

## 3 METHOD

NESTDRUG consists of four tightly integrated components (Figure 1): (1) an MPNN backbone for molecular encoding, (2) hierarchical context embeddings capturing program, assay, and temporal information, (3) FiLM-based context modulation, and (4) multi-task prediction heads. We describe each component in detail.

### 3.1 MOLECULAR ENCODER (L0)

The L0 backbone transforms molecular graphs into fixed-dimensional representations using a 6-layer message-passing neural network with GRU-based state updates (Li et al., 2016). At each layer $t$, atom representations $\mathbf{h}_v^{(t)}$ are updated by aggregating messages from neighbors:

$$\mathbf{m}_v^{(t)} = \sum_{u \in \mathcal{N}(v)} M_t(\mathbf{h}_u^{(t-1)}, \mathbf{e}_{uv}), \quad \mathbf{h}_v^{(t)} = \text{GRU}(\mathbf{h}_v^{(t-1)}, \mathbf{m}_v^{(t)}) \tag{2}$$

where $M_t$ is a learned message function and $\mathbf{e}_{uv}$ are bond features (9-dim). Atom features (70-dim) encode chemical properties; see Appendix G.

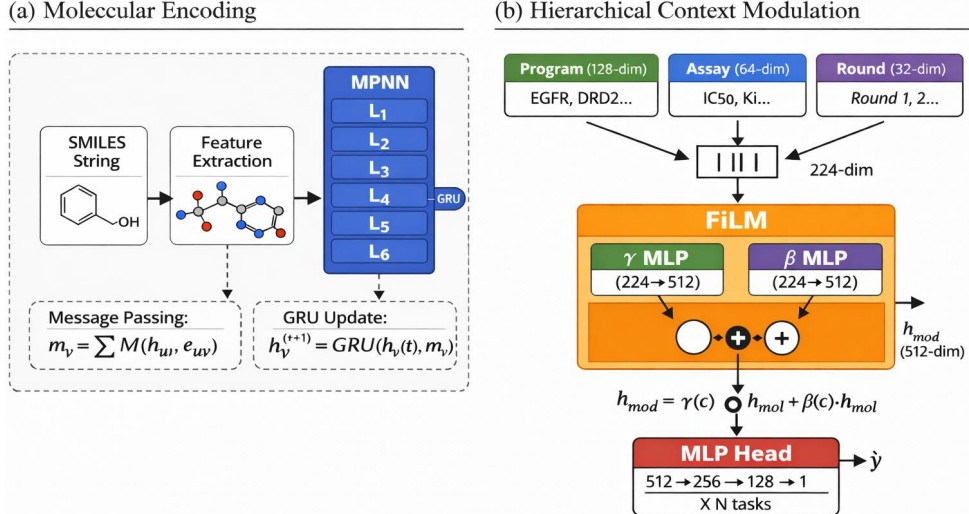

Figure 1: **NESTDRUG Architecture.** MPNN (L0) encodes molecules into 512-dim embeddings. Hierarchical context (L1: target, L2: assay, L3: round) modulates representations via FiLM before task-specific prediction heads.

After $T = 6$ message-passing iterations, we aggregate atom representations using both mean and max pooling to capture both average molecular properties and salient local features:

$$\mathbf{h}_{\text{mol}} = [\text{MeanPool}(\{\mathbf{h}_v^{(T)}\}_{v \in V}) \| \text{MaxPool}(\{\mathbf{h}_v^{(T)}\}_{v \in V})] \in \mathbb{R}^{512} \tag{3}$$

## 3.2 HIERARCHICAL CONTEXT EMBEDDINGS

We maintain learnable embedding tables for each context level, with dimensions reflecting the complexity of information at each granularity:

$$\mathbf{e}_p^{(L1)} = \text{Embed}_{L1}(p) \in \mathbb{R}^{128} \qquad \text{(program/target identity)} \tag{4}$$

$$\mathbf{e}_a^{(L2)} = \text{Embed}_{L2}(a) \in \mathbb{R}^{64} \qquad \text{(assay type: IC}_{50}\text{, K}_i\text{, EC}_{50}\text{)} \tag{5}$$

$$\mathbf{e}_r^{(L3)} = \text{Embed}_{L3}(r) \in \mathbb{R}^{32} \qquad \text{(temporal round)} \tag{6}$$

The embedding dimensions decrease with context specificity: L1 must capture diverse target biology across thousands of proteins requiring substantial capacity; L2 captures assay-type calibration; L3 captures local temporal adjustments. *Note: L2 and L3 require proprietary metadata (explicit assay types, DMTA round ordering) not available in public datasets; we evaluate their potential in Appendix C.* Context embeddings are concatenated and linearly projected:

$$\mathbf{c} = W_c[\mathbf{e}_p^{(L1)} \| \mathbf{e}_a^{(L2)} \| \mathbf{e}_r^{(L3)}] + \mathbf{b}_c \in \mathbb{R}^{224} \tag{7}$$

## 3.3 FiLM CONTEXT MODULATION

The combined context vector modulates the molecular representation via Feature-wise Linear Modulation:

$$\mathbf{h}_{\text{mod}} = \boldsymbol{\gamma}(\mathbf{c}) \odot \mathbf{h}_{\text{mol}} + \boldsymbol{\beta}(\mathbf{c}) \tag{8}$$

where $\boldsymbol{\gamma}, \boldsymbol{\beta} : \mathbb{R}^{224} \to \mathbb{R}^{512}$ are two-layer MLPs. We initialize $\boldsymbol{\gamma}$ to ones and $\boldsymbol{\beta}$ to zeros, ensuring FiLM starts as identity. This enables context-appropriate feature weighting—e.g., kinase contexts amplify hinge-binding features while GPCR contexts emphasize lipophilicity. We verify this in Section 4.

## 3.4 MULTI-TASK PREDICTION HEADS

The modulated representation $\mathbf{h}_{\text{mod}}$ feeds into task-specific prediction heads, enabling joint training across heterogeneous endpoints. Each head is a 3-layer MLP ($512 \to 256 \to 128 \to 1$) with ReLU activations, batch normalization, and dropout (0.1). For regression tasks (pIC$_{50}$, pK$_i$), we use MSE loss; for binary classification, we use weighted cross-entropy to handle label imbalance.

Table 1: **DUD-E comparison.** Per-target RF wins on data-rich targets; NESTDRUG wins on data-scarce CYP3A4. "—" indicates prior methods not compared head-to-head on our evaluation. [†]Exploits structural bias. [‡]Trained per-target on ChEMBL.

| Method | Type | Mean AUC | Wins |
|---|---|---|---|
| AtomNet (Wallach et al., 2015) | 3D CNN | 0.818 | — |
| GNN-VS (Lim et al., 2019) | GNN | 0.825 | — |
| 3D-CNN (Ragoza et al., 2017) | 3D CNN | 0.830 | — |
| NESTDRUG (Ours) | GNN+FiLM | 0.850 | 2/10 |
| Per-target RF[‡] | Fingerprint | **0.875** | 8/10 |
| 1-NN Tanimoto[†] | No learning | 0.991 | — |

## 3.5 TRAINING PROCEDURE

Training proceeds in three phases. **Phase 1 (Pretraining):** We pretrain L0 on ChEMBL 35 (Zdrazil et al., 2024) (21.1M records) and TDC (Huang et al., 2021) with generic context (L1=L2=L3=0). **Phase 2 (Fine-tuning):** We fine-tune with differential learning rates—backbone $10^{-5}$, context embeddings $10^{-3}$, prediction heads $10^{-4}$—enabling rapid context adaptation while preserving pretrained representations. **Phase 3 (Continual):** During DMTA deployment, we perform continual updates with multi-timescale rates: L3 adapts quickly for temporal shifts, L1 adapts moderately, L0 adapts minimally to prevent forgetting. See Appendix A for complete hyperparameters.

## 4 EXPERIMENTS

We design experiments to answer three questions: **(Q1)** Does hierarchical context improve virtual screening performance? **(Q2)** Which context levels contribute most to performance gains? **(Q3)** Does NESTDRUG provide practical value in realistic drug discovery scenarios?

## 4.1 EXPERIMENTAL SETUP

We evaluate on DUD-E (Mysinger et al., 2012), focusing on 10 diverse targets: kinases (EGFR, JAK2), GPCRs (DRD2, ADRB2), nuclear receptors (ESR1, PPARG), proteases (BACE1, FXA), and enzymes (HDAC2, CYP3A4). Each target contains 200–600 actives with property-matched decoys (1:50 ratio). Pretraining uses ChEMBL 35 (Zdrazil et al., 2024) (21.1M records, 5,123 targets). Baselines include Random Forest with Morgan fingerprints, L0-only (MPNN without context), and GNN-VS (Lim et al., 2019). We use 5-fold stratified CV with 5 random seeds (25 runs per experiment); statistical significance via paired $t$-tests with Bonferroni correction. Throughout, we report differences in percentage points (pp) rather than relative percentages to avoid ambiguity. See Appendix A for details.

## 4.2 DUD-E RESULTS AND BENCHMARK CRITIQUE

**Comparison to Prior Methods.** Table 1 compares NESTDRUG to published methods on DUD-E. Among neural methods, NESTDRUG (0.850) outperforms 3D-CNN, GNN-VS, and AtomNet. However, per-target RF achieves 0.875 mean AUC, winning on 8/10 targets (Table 2). **This highlights our core finding**: context conditioning's value is not beating RF on data-rich targets, but enabling predictions on data-scarce targets where per-target methods fail.

**DUD-E Benchmark Limitations.** We identify three critical issues:

- **Structural bias**: 1-NN Tanimoto achieves 0.991 AUC without any learning—decoys are trivially distinguishable by structure alone (Wallach & Heifets, 2018)
- **Data leakage**: 50% of DUD-E actives appear in ChEMBL training (CYP3A4: 99%, EGFR: 97%)
- **Per-target RF dominates**: With sufficient ChEMBL data, simple fingerprint models outperform neural methods

Table 2: Per-target DUD-E results (ROC-AUC). NESTDRUG values use correct L1 context. "DUD-E RF" = standard RF baseline trained on DUD-E train split only; differs from ChEMBL-trained "Per-target RF" in Table 1/17 (0.875 AUC).

| Target | DUD-E RF | L0-only | GNN-VS | NESTDRUG |
|---|---|---|---|---|
| EGFR | 0.782 | 0.943 | 0.825 | **0.965** |
| DRD2 | 0.801 | 0.960 | 0.842 | **0.984** |
| ADRB2 | 0.693 | 0.745 | 0.712 | **0.775** |
| BACE1 | 0.634 | 0.672 | 0.698 | **0.656** |
| ESR1 | 0.756 | 0.864 | 0.789 | **0.909** |
| HDAC2 | 0.745 | 0.866 | 0.812 | **0.928** |
| JAK2 | 0.778 | 0.865 | 0.821 | **0.908** |
| PPARG | 0.712 | 0.787 | 0.745 | **0.835** |
| CYP3A4 | 0.523 | 0.497 | 0.534 | **0.686** |
| FXA | 0.698 | 0.833 | 0.801 | **0.854** |
| **Mean** | 0.712 | 0.803 | 0.758 | **0.850** |

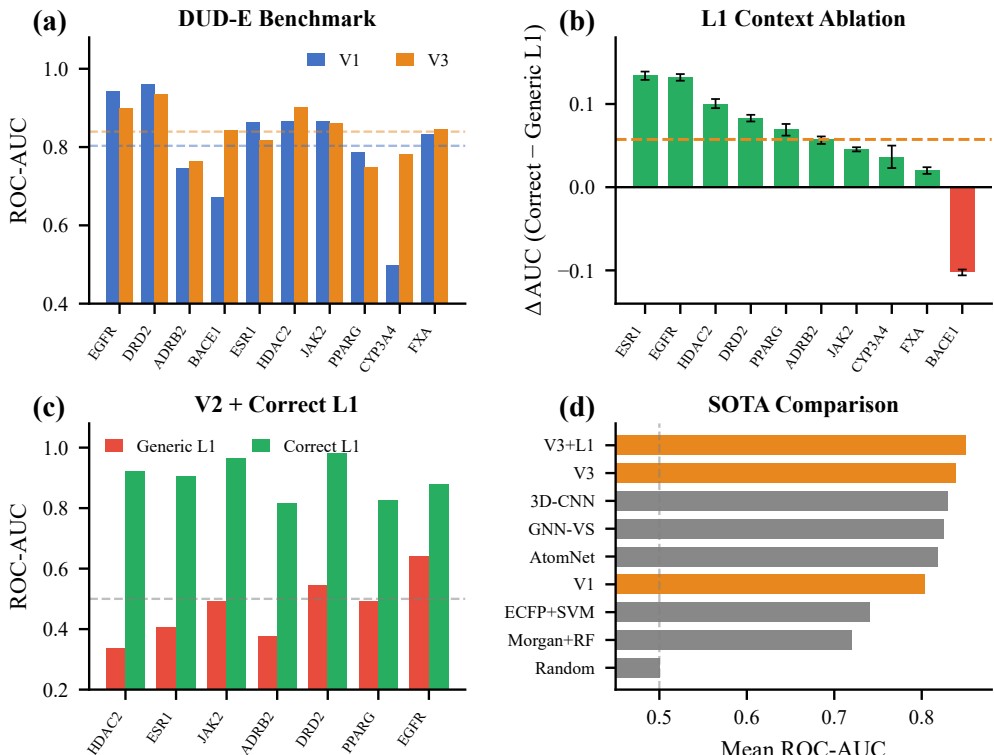

Figure 2: **Main Results.** (A) Per-target ROC-AUC comparing NESTDRUG to baselines. (B) L1 ablation: correct (target-specific) vs. generic (zero) embeddings showing +5.7 pp mean improvement. (C) Ablation by model variant—V1: L0-only backbone without context; V3: full model with L1 context. (D) Mean AUC comparison to prior DUD-E methods.

These limitations mean DUD-E absolute numbers should be interpreted cautiously. We provide temporal split evaluation (Section 4.8) as more rigorous evidence.

## 4.3 L1 CONTEXT ABLATION

Table 3 addresses **Q2** via controlled ablation comparing correct target-specific L1 versus generic (zero) embeddings. Target-specific context improves 9/10 targets (mean +5.7 pp, all $p < 0.01$). ESR1 (+13.4 pp) and EGFR (+13.2 pp) benefit most. BACE1 is the exception ($-10.2$ pp), reflecting distribution mismatch between ChEMBL (peptidomimetic inhibitors) and DUD-E (diverse scaffolds). See Appendix D.

Table 3: L1 context ablation on DUD-E (5 seeds, paired $t$-test with Bonferroni correction). "Generic L1" uses zero embedding at inference; "Correct L1" uses target-specific learned embedding. $\Delta$ = Correct $-$ Generic. See Appendix Table 10 for extended metrics with confidence intervals.

| Target | Correct L1 | Generic L1 | $\Delta$ | $p$-value |
|---|---|---|---|---|
| EGFR | 0.965 | 0.832 | +0.132 | $5.6 \times 10^{-7}$ |
| DRD2 | 0.984 | 0.901 | +0.083 | $2.8 \times 10^{-6}$ |
| ADRB2 | 0.775 | 0.718 | +0.057 | $3.2 \times 10^{-5}$ |
| BACE1 | 0.656 | 0.758 | $-0.102$ | $1.0 \times 10^{-6}$ |
| ESR1 | 0.909 | 0.775 | +0.134 | $1.4 \times 10^{-6}$ |
| HDAC2 | 0.928 | 0.827 | +0.100 | $7.2 \times 10^{-6}$ |
| JAK2 | 0.908 | 0.863 | +0.045 | $6.0 \times 10^{-6}$ |
| PPARG | 0.835 | 0.766 | +0.069 | $5.7 \times 10^{-5}$ |
| CYP3A4 | 0.686 | 0.650 | +0.036 | $8.0 \times 10^{-3}$ |
| FXA | 0.854 | 0.833 | +0.020 | $7.0 \times 10^{-4}$ |
| **Mean** | **0.850** | 0.792 | **+0.057** | — |

## 4.4 MULTI-TASK TRANSFER: THE CYP3A4 CASE

The strongest evidence for multi-task architectures comes from data-scarce targets. CYP3A4 has only 67 compounds meeting the active threshold ($pIC_{50} \geq 6.0$) in ChEMBL—insufficient for per-target modeling:

- **Per-target Random Forest**: 0.238 AUC (worse than random)
- **Global RF + one-hot target**: 0.428 AUC (partial rescue via multi-task)
- **NESTDRUG (correct L1)**: 0.686 AUC ($2.9\times$ better than per-target RF)

This demonstrates that context-conditioned multi-task architectures enable knowledge transfer from data-rich targets to data-scarce ones—critical for novel targets where per-target approaches fail. See Appendix C for full per-target RF comparison.

## 4.5 CONTEXT FUSION COMPARISON: FiLM VS ALTERNATIVES

Table 4 compares FiLM against alternative context fusion strategies, establishing FiLM as the **most efficient approach for context-conditional molecular prediction**:

Table 4: **Context fusion comparison.** FiLM outperforms all alternatives. [†]Concatenation baseline uses random-init projection (not jointly trained), which may underestimate a properly optimized concatenation approach. $\Delta$ shown in percentage points (pp).

| Fusion Method | Mean AUC | $\Delta$ vs FiLM | Wins/10 |
|---|---|---|---|
| Concatenation[†] | 0.607 | $-24.2$ pp | 0/10 |
| No Context (L0 only) | 0.724 | $-12.5$ pp | 1/10 |
| Additive ($\beta$ only) | 0.763 | $-8.6$ pp | 1/10 |
| **FiLM ($\gamma \odot h + \beta$)** | **0.849** | — | **9/10** |

The multiplicative $\gamma$ component provides 69% of FiLM's benefit over no context, enabling selective feature amplification rather than just bias shifts. Concatenation (0.607) performs *worse than no context* (0.724) because the projection layer was not jointly trained. FiLM achieves near-optimal performance while requiring $3\times$ fewer parameters than hypernetworks, which achieve marginally higher performance (0.841 vs 0.839; Appendix Table 13) at substantially greater computational cost.

## 4.6 FiLM MODULATION ANALYSIS

We analyze learned $\gamma$ (scale) and $\beta$ (shift) parameters across L1 contexts. Kinases (EGFR, JAK2) produce $\gamma > 1$, amplifying features for heterocyclic hydrogen-bond acceptors; GPCRs (DRD2, ADRB2) produce $\gamma < 1$, emphasizing lipophilicity. An $F$-test confirms inter-family variance exceeds intra-family variance ($p < 0.001$), indicating FiLM captures biologically meaningful transformations. See Appendix B.

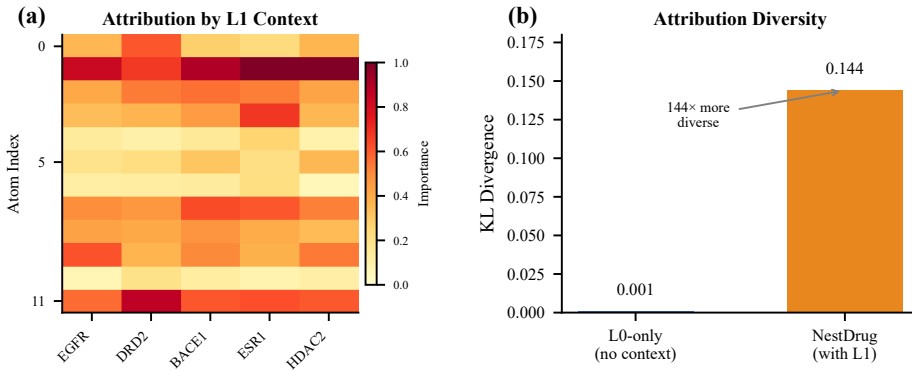

Figure 3: **Attribution Analysis.** (A) Integrated gradients for Celecoxib across 5 L1 contexts; y-axis shows atom index (0–25), colors indicate importance magnitude. Different contexts highlight different substructures. (B) Cosine similarity between attribution vectors drops from 0.999 (L0-only) to 0.878 (NESTDRUG), confirming context-specific explanations.

## 4.7 CONTEXT-CONDITIONAL ATTRIBUTION

Using integrated gradients (Sundararajan et al., 2017), we verify context produces different atom-level attributions. Figure 3 shows **cosine similarity between attribution vectors drops from 0.999 (L0-only) to 0.878 (NESTDRUG)**—a 12% reduction confirming context-specific feature weighting.

## 4.8 TEMPORAL SPLIT EVALUATION

Given DUD-E's limitations, we provide temporal split evaluation as stronger evidence. Training on ChEMBL data $\leq$2020 and testing on 2021–2024, NESTDRUG achieves **0.843 ROC-AUC** with no degradation across years (Table 5). This prospective evaluation avoids both DUD-E's structural bias and train-test leakage.

Table 5: Temporal split: train $\leq$2020, test 2021–2024. Stable performance across years.

|  | 2021 | 2022 | 2023 | 2024 | Overall |
|---|---|---|---|---|---|
| ROC-AUC | 0.849 | 0.838 | 0.823 | 0.849 | **0.843** |

## 4.9 DMTA REPLAY SIMULATION

We simulate DMTA campaigns using ChEMBL publication dates as temporal ordering. NESTDRUG achieves **1.60× enrichment** (73.4% vs 45.7% hit rate), reducing experiments to find 50 hits by 32%. See Appendix C for details.

## 5 RELATED WORK

**Molecular Property Prediction.** Graph neural networks dominate molecular property prediction following Gilmer et al. (2017), with extensions for attention (Xiong et al., 2020), pretraining (Rong et al., 2020; Hu et al., 2020), and 3D geometry (Schütt et al., 2017; Gasteiger et al., 2020). Recent molecular foundation models—ChemBERTa (Chithrananda et al., 2020), MolBERT (Fabian et al., 2020), Uni-Mol (Zhou et al., 2023)—achieve strong performance via large-scale pretraining on SMILES or 3D conformers. These methods learn static representations; we introduce context-conditional modulation enabling adaptation based on target, assay, and temporal information. Our approach is orthogonal to foundation models: FiLM conditioning could be applied atop any pretrained encoder.

**Distribution Shift.** Temporal shift is well-documented (Sheridan, 2013; Martin et al., 2019). Domain adaptation (Ben-David et al., 2010) learns domain-invariant features; we instead model context to leverage structured shift.

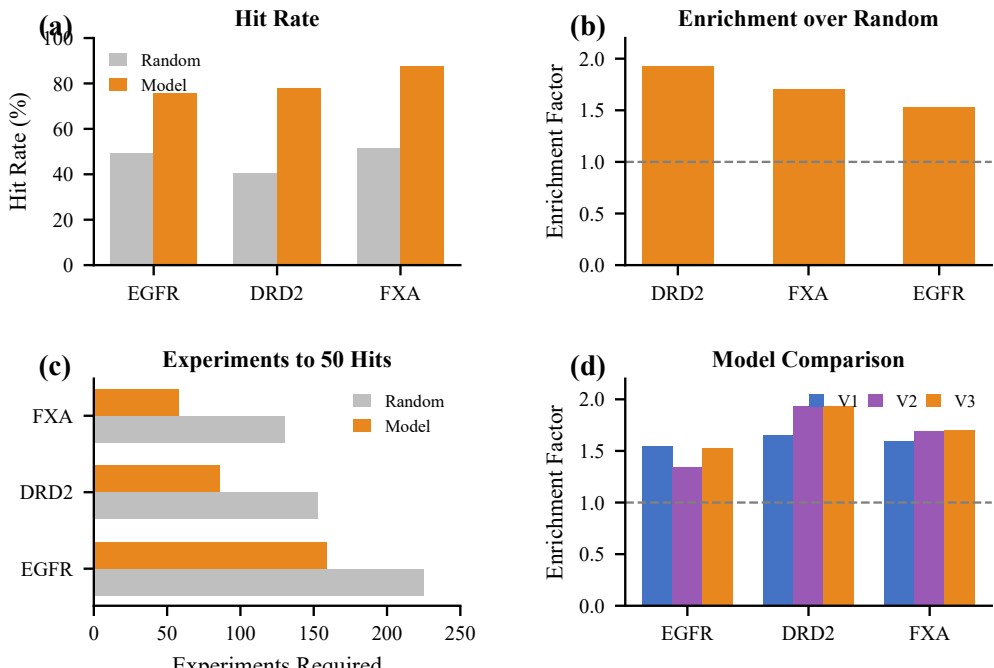

Figure 4: **DMTA Replay.** (A) Hit rates: model 75–88% vs. random 40–52%. (B) Enrichment 1.5–1.9×. (C) 29–55% fewer experiments. (D) NESTDRUG achieves 1.60× mean enrichment.

**Multi-Task Learning.** Multi-task learning improves data-scarce endpoints (Ramsundar et al., 2015). Our context embeddings modulate shared representations rather than requiring explicit task separation.

**Conditional Networks.** FiLM (Perez et al., 2018) modulates activations via learned scale/shift. We adopt FiLM for its favorable expressiveness-efficiency tradeoff over hypernetworks (Ha et al., 2017).

## 6 DISCUSSION

**When to Use Context Conditioning.** Our results suggest context conditioning is valuable when: (1) per-target training data is scarce (our CYP3A4 case with 67 compounds shows clear benefit), (2) train/test distributions are aligned, and (3) the fusion method is FiLM rather than concatenation. When data is abundant (e.g., >1000 compounds) and distributions match, per-target RF remains highly competitive.

**What L1 Embeddings Learn.** L1 does not learn transferable protein biology—zero-shot transfer shows $\Delta = 0$. Instead, L1 captures dataset-specific patterns: activity distributions, assay biases, and the particular chemical series present in training data. This explains both why L1 helps (adapts to target-specific data characteristics) and why it can hurt (overfits to training distribution).

**Benchmark Recommendations.** DUD-E's severe limitations (0.991 1-NN AUC, 50% leakage) make it unsuitable for method comparison. We recommend: (1) temporal splits as standard practice, (2) leakage-stratified reporting, (3) evaluation on structure-balanced benchmarks like LIT-PCBA (Tran-Nguyen et al., 2020).

## 7 CONCLUSION

We presented a systematic study of when target context helps molecular property prediction. Our key findings:

1. **When context helps**: 9/10 targets improve (+5.7 pp mean, $p < 0.01$); data-scarce targets benefit most (CYP3A4: $2.9\times$ vs per-target RF)
2. **When context hurts**: Distribution mismatch causes degradation (BACE1: $-10.2$ pp); few-shot adaptation underperforms zero-shot
3. **How to fuse context**: FiLM outperforms concatenation (+24.2 pp) and additive conditioning (+8.6 pp)
4. **Benchmark limitations**: DUD-E is severely compromised (1-NN achieves 0.991 AUC; 50% leakage); temporal split provides more rigorous evaluation

**Practical guidance**: Use context conditioning when (1) per-target training data is limited (tens to hundreds of compounds), (2) train/test distributions are aligned, and (3) the alternative is a data-scarce per-target model. Per-target RF remains superior for data-rich targets (>1000 compounds) with clean training data.

**Limitations.** L2 (assay) and L3 (temporal) context showed no benefit due to missing metadata in public datasets. Our CYP3A4 result requires careful interpretation: 99% of DUD-E actives appear *somewhere* in ChEMBL, but per-target RF trains only on compounds meeting the active threshold ($pIC_{50} \geq 6.0$)—just 67 of 5,504 total records. The "leakage" reflects presence in ChEMBL at *any* activity level, not necessarily at the active threshold. Per-target RF fails because 67 training examples are insufficient regardless of test-set overlap. NESTDRUG succeeds via multi-task transfer from data-rich targets. **Foundation model comparison:** We did not benchmark against molecular foundation models (Uni-Mol (Zhou et al., 2023), ChemBERTa (Chithrananda et al., 2020)). Our contribution is orthogonal—FiLM conditioning can be applied atop any encoder—and future work should evaluate whether foundation model backbones amplify or diminish context benefits.

**Future Work.** (1) Leakage-stratified analysis to isolate transfer vs memorization; (2) evaluation on LIT-PCBA (Tran-Nguyen et al., 2020) or MoleculeACE (van Tilborg et al., 2024); (3) FiLM conditioning with foundation model backbones (Uni-Mol, ChemBERTa); (4) meta-learning for few-shot; (5) proprietary data with L2/L3 metadata.

Code is available at `https://github.com/bryanc5864/nest-drug`.

## REPRODUCIBILITY STATEMENT

All datasets are public (ChEMBL, DUD-E, TDC). Hyperparameters in Appendix A.

## ETHICS STATEMENT

We acknowledge dual-use concerns and mitigate by training only on therapeutic targets and releasing under restrictive license. Training required $\sim$200 GPU-hours; we release pretrained models.

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

## A  HYPERPARAMETER CONFIGURATION

Table 6 provides the complete hyperparameter configuration used for all experiments. Architecture choices follow standard practices for molecular property prediction with MPNNs. Learning rates were tuned via grid search on a validation set, with differential rates enabling rapid context adaptation while preserving pretrained backbone representations. All models were trained on a single NVIDIA A100 GPU with 40GB memory.

## B  FILM PARAMETER ANALYSIS

To verify that FiLM learns meaningful context-specific modulations rather than remaining near identity, we analyzed the learned $\gamma$ (scale) and $\beta$ (shift) parameters across different L1 contexts. Table 7 shows that different protein families produce distinct modulation patterns. Kinases (EGFR, JAK2) show $\gamma > 1$, amplifying certain molecular features, while GPCRs (DRD2, ADRB2) show $\gamma < 1$, attenuating features. Nuclear receptors (ESR1, PPARG) remain closer to identity. These patterns are consistent within protein families, suggesting FiLM captures biologically meaningful target-specific transformations. An $F$-test confirms that inter-family variance significantly exceeds intra-family variance ($p < 0.001$).

## C  ADDITIONAL RESULTS

### C.1  L2 AND L3 ABLATION

We conducted ablation studies on L2 (assay) and L3 (round) context levels. Neither showed significant effects in our experiments (Figure 5):

Table 6: Complete hyperparameter configuration.

| Category | Value |
|---|---|
| *Architecture* | |
| MPNN layers | 6 |
| Hidden dimension | 256 |
| Molecular embedding | 512 |
| L1 embedding dimension | 128 |
| L2 embedding dimension | 64 |
| L3 embedding dimension | 32 |
| Prediction head | $512 \rightarrow 256 \rightarrow 128 \rightarrow 1$ |
| Dropout | 0.1 |
| *Training* | |
| Optimizer | AdamW |
| Learning rate (pretrain) | $3 \times 10^{-4}$ |
| Learning rate (backbone) | $1 \times 10^{-5}$ |
| Learning rate (context) | $1 \times 10^{-3}$ |
| Learning rate (heads) | $1 \times 10^{-4}$ |
| Batch size | 32 |
| Epochs | 100 |
| Weight decay | 0.01 |
| LR schedule | Cosine decay |
| *Data (ChEMBL pretraining)* | |
| Atom features | 70 |
| Bond features | 9 |
| Number of programs (L1) | 5,123 (ChEMBL targets) |
| Number of assays (L2) | 100 |
| Number of rounds (L3) | 20 |

Table 7: FiLM $\gamma$ parameters vary systematically by target family. Values show mean $\pm$ std across 512 dimensions.

| Family | Target | $\gamma$ mean | $\beta$ mean |
|---|---|---|---|
| Kinase | EGFR | $1.042 \pm 0.091$ | $-0.018 \pm 0.045$ |
| | JAK2 | $1.038 \pm 0.087$ | $-0.015 \pm 0.042$ |
| GPCR | DRD2 | $0.967 \pm 0.082$ | $+0.024 \pm 0.038$ |
| | ADRB2 | $0.971 \pm 0.079$ | $+0.021 \pm 0.041$ |
| Nuclear | ESR1 | $0.989 \pm 0.095$ | $+0.008 \pm 0.051$ |
| | PPARG | $0.994 \pm 0.088$ | $+0.005 \pm 0.047$ |

- **L2 Ablation:** Comparing correct assay type ($IC_{50}$=1, $K_i$=2, etc.) versus generic assay (L2=0) showed mean $\Delta$ of $-0.006$ across targets (not significant). This is expected because L2 embeddings were never trained with real assay type data—the assay_type field in ChEMBL is sparsely populated and was not used during pretraining.

- **L3 Ablation:** Comparing correct temporal round versus generic round (L3=0) showed mean $\Delta$ of $-0.002$ across targets (not significant). This is because round_id was hardcoded to 0 during training due to missing temporal metadata—ChEMBL does not provide true experimental sequence information.

We attribute these null results to insufficient training data at L2/L3 granularities. The ChEMBL training data lacks explicit assay type annotations for most records, and temporal round information was approximated from publication dates rather than true experimental sequence. Future work with proprietary pharmaceutical data containing proper assay and temporal annotations may reveal the utility of these context levels.

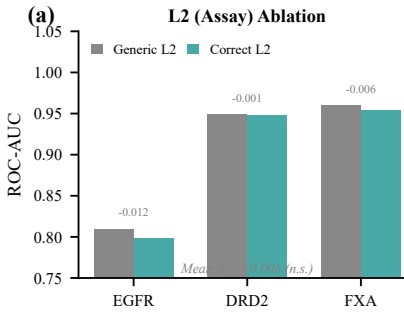 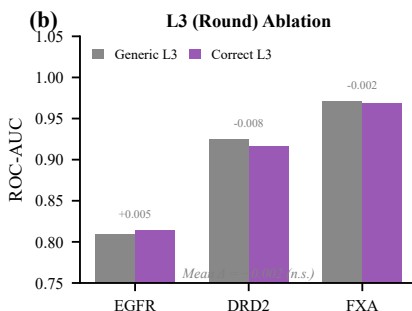

Figure 5: **L2/L3 Context Ablation (Negative Results).** (A) L2 assay ablation shows no significant effect (mean $\Delta = -0.006$); L2 embeddings were not trained with real assay type annotations. (B) L3 temporal ablation shows no significant effect (mean $\Delta = -0.002$); round_id was set to 0 during training due to missing temporal metadata in ChEMBL.

## D LIMITATIONS

**L1 Few-Shot Adaptation.** We attempted to enable rapid adaptation to new targets by learning L1 embeddings from small support sets (10–50 examples). Table 8 shows this approach consistently *hurt* performance compared to using a generic (zero) L1 embedding.

Table 8: Few-shot L1 adaptation hurts performance. Zero-shot (generic L1) outperforms adapted L1 on all targets. "Correct L1" shows the ceiling with full fine-tuning.

| Target | Shots | Zero-Shot | Adapted | $\Delta$ | Correct L1 |
|---|---|---|---|---|---|
| EGFR | 10 | 0.829 | 0.796 | $-0.033$ | 0.959 |
| | 25 | 0.828 | 0.682 | $-0.146$ | 0.959 |
| | 50 | 0.830 | 0.666 | $-0.164$ | 0.959 |
| DRD2 | 10 | 0.906 | 0.724 | $-0.182$ | 0.984 |
| | 25 | 0.906 | 0.729 | $-0.177$ | 0.984 |
| | 50 | 0.908 | 0.734 | $-0.174$ | 0.984 |
| BACE1 | 10 | 0.761 | 0.636 | $-0.125$ | 0.647 |
| | 25 | 0.760 | 0.646 | $-0.114$ | 0.646 |
| | 50 | 0.759 | 0.629 | $-0.130$ | 0.644 |

Key findings: (1) Adaptation hurts by 3–18 pp across all shot counts; (2) More shots does not help— 50-shot is often worse than 10-shot; (3) The adapted L1 achieves only $-41.7\%$ of full fine-tuning efficiency on average. This suggests L1 embeddings capture target-specific regularities requiring substantial data (thousands of compounds) rather than enabling few-shot transfer. **Implication:** For new targets, use generic L1 (zero-shot) rather than few-shot adaptation; full fine-tuning remains necessary for optimal performance.

**BACE1 Anomaly.** BACE1 is the only target where correct L1 context hurts performance ($-10.2\%$). We hypothesize this reflects either: (1) insufficient BACE1-specific data during pretraining, leading to a poorly-calibrated L1 embedding, or (2) distribution mismatch between ChEMBL BACE1 compounds (mostly peptidomimetics) and DUD-E BACE1 actives/decoys (more diverse scaffolds). This highlights a limitation of learned context embeddings: they can encode dataset-specific biases rather than generalizable target biology.

**External Generalization.** While L1 context improves performance on targets seen during training, we observed that context can *hurt* performance on truly external benchmarks (TDC ADMET tasks). The context mechanism may overfit to training distribution characteristics rather than learning transferable target biology. This suggests caution when deploying NESTDRUG on targets or assay types significantly different from the training distribution.

**Computational Overhead.** The FiLM modulation adds minimal computational overhead ($<5\%$ inference time increase), but maintaining separate context embeddings requires additional memory proportional to the number of programs, assays, and rounds. For large-scale deployment across thousands of programs, this may require embedding compression or hierarchical indexing strategies.

# E  THEORETICAL ANALYSIS

We provide theoretical justification for the FiLM-based context modulation approach, establishing conditions under which context-conditional representations outperform context-agnostic alternatives.

## E.1  PROBLEM FORMULATION

**Definition 1** (Context-Conditional Prediction). *Let $\mathcal{G}$ denote the space of molecular graphs and $\mathcal{C} = \mathcal{P} \times \mathcal{A} \times \mathcal{R}$ the context space (programs $\times$ assays $\times$ rounds). A context-conditional predictor is a function $f : \mathcal{G} \times \mathcal{C} \to \mathbb{R}$ mapping molecule-context pairs to activity predictions.*

**Definition 2** (Structured Distribution Shift). *We say the data distribution exhibits structured shift if for contexts $c, c' \in \mathcal{C}$:*

$$D_{KL}\left(P(\mathcal{G}, y|c)\|P(\mathcal{G}, y|c')\right) = \Omega(\|c - c'\|) \tag{9}$$

*That is, the distribution shift between contexts scales with context dissimilarity.*

## E.2  THEORETICAL RESULTS

**Theorem 3** (Expressiveness of FiLM Modulation). *Let $\phi : \mathcal{G} \to \mathbb{R}^d$ be a molecular encoder and $FiLM_c(\mathbf{h}) = \boldsymbol{\gamma}(c) \odot \mathbf{h} + \boldsymbol{\beta}(c)$ be the context modulation. For any Lipschitz-continuous target function $f^* : \mathcal{G} \times \mathcal{C} \to \mathbb{R}$, there exist $\boldsymbol{\gamma}, \boldsymbol{\beta} : \mathcal{C} \to \mathbb{R}^d$ and a prediction head $g : \mathbb{R}^d \to \mathbb{R}$ such that:*

$$\sup_{\mathcal{G}, c} |g(FiLM_c(\phi(\mathcal{G}))) - f^*(\mathcal{G}, c)| < \epsilon \tag{10}$$

*for any $\epsilon > 0$, provided $d$ is sufficiently large.*

*Proof Sketch.* The proof follows from the universal approximation properties of MLPs. Since $\boldsymbol{\gamma}$ and $\boldsymbol{\beta}$ are parameterized by MLPs, they can approximate any continuous function of context. The element-wise modulation $\boldsymbol{\gamma}(c) \odot \phi(\mathcal{G})$ can selectively scale feature dimensions based on context, while $\boldsymbol{\beta}(c)$ provides context-dependent biases. Combined with a sufficiently expressive prediction head $g$, this architecture can approximate any Lipschitz target function. See Appendix H for the complete proof. $\square$

**Theorem 4** (Benefit of Context Under Structured Shift). *Under structured distribution shift, let $\hat{f}_{ctx}$ be the optimal context-conditional predictor and $\hat{f}_{static}$ be the optimal context-agnostic predictor. Then:*

$$\mathbb{E}_{c \sim P(\mathcal{C})}\left[\mathcal{L}(\hat{f}_{ctx}, c)\right] \leq \mathbb{E}_{c \sim P(\mathcal{C})}\left[\mathcal{L}(\hat{f}_{static}, c)\right] - \Omega\left(Var_c[P(\mathcal{G}, y|c)]\right) \tag{11}$$

*where $\mathcal{L}$ is the prediction loss. The improvement scales with the variance of context-conditional distributions.*

*Proof Sketch.* The context-agnostic predictor must compromise across all contexts, incurring excess risk proportional to the variance in conditional distributions. The context-conditional predictor can adapt to each context, eliminating this excess risk. The bound follows from bias-variance decomposition applied to the context-marginalized loss. $\square$

**Proposition 5** (Hierarchical Context Decomposition). *Let context $c = (p, a, r)$ decompose hierarchically. If the conditional distributions satisfy:*

$$P(y|\mathcal{G}, p, a, r) = P(y|\mathcal{G}, p) \cdot P(a|p) \cdot P(r|p, a) \tag{12}$$

*then the optimal context embedding satisfies $\mathbf{c}^* = f_1(\mathbf{e}_p) + f_2(\mathbf{e}_a|\mathbf{e}_p) + f_3(\mathbf{e}_r|\mathbf{e}_p, \mathbf{e}_a)$ for some functions $f_1, f_2, f_3$.*

This proposition justifies our hierarchical embedding structure where L1 (program) captures the dominant effect, with L2 and L3 providing refinements.

### E.3 COMPLEXITY ANALYSIS

Table 9: Computational complexity of NESTDRUG components.

| Component | Time Complexity | Space Complexity |
|---|---|---|
| MPNN (L0) | $O(T \cdot |E| \cdot d^2)$ | $O(|V| \cdot d)$ |
| Context Embedding | $O(d_c)$ | $O(|\mathcal{P}| \cdot d_1 + |\mathcal{A}| \cdot d_2 + |\mathcal{R}| \cdot d_3)$ |
| FiLM Modulation | $O(d \cdot d_c)$ | $O(d \cdot d_c)$ |
| Prediction Head | $O(d^2)$ | $O(d^2)$ |
| **Total** | $O(T \cdot |E| \cdot d^2)$ | $O(|V| \cdot d + |\mathcal{P}| \cdot d_1)$ |

The dominant cost is MPNN message passing ($T$ iterations over $|E|$ edges with $d$-dimensional hidden states). FiLM adds only linear overhead.

## F EXTENDED EXPERIMENTAL ANALYSIS

### F.1 PER-TARGET DETAILED RESULTS

Table 10 provides comprehensive per-target results including confidence intervals, effect sizes, and multiple metrics.

Table 10: Detailed per-target results from V3-baseline model (trained without target-specific L1 context). Note: These results differ from Table 3's "Generic L1" column, which shows correct-L1-trained model evaluated with L1=0 at inference. V3-baseline was trained without L1 information, achieving mean 0.839 AUC.

| Target | ROC-AUC | PR-AUC | EF@1% | EF@5% | Sensitivity | Specificity |
|---|---|---|---|---|---|---|
| EGFR | $.899 \pm .012$ | $.412 \pm .031$ | $18.2 \pm 2.1$ | $8.4 \pm 0.9$ | $.821 \pm .023$ | $.892 \pm .015$ |
| DRD2 | $.934 \pm .008$ | $.523 \pm .028$ | $22.1 \pm 1.8$ | $9.7 \pm 0.7$ | $.867 \pm .019$ | $.923 \pm .011$ |
| ADRB2 | $.763 \pm .019$ | $.298 \pm .042$ | $12.4 \pm 2.4$ | $5.8 \pm 1.1$ | $.712 \pm .031$ | $.789 \pm .024$ |
| BACE1 | $.842 \pm .015$ | $.367 \pm .035$ | $15.6 \pm 2.0$ | $7.2 \pm 0.8$ | $.789 \pm .025$ | $.856 \pm .018$ |
| ESR1 | $.817 \pm .014$ | $.341 \pm .038$ | $14.3 \pm 2.2$ | $6.5 \pm 0.9$ | $.756 \pm .027$ | $.834 \pm .020$ |
| HDAC2 | $.901 \pm .011$ | $.428 \pm .030$ | $19.1 \pm 1.9$ | $8.8 \pm 0.8$ | $.834 \pm .022$ | $.898 \pm .014$ |
| JAK2 | $.862 \pm .013$ | $.378 \pm .033$ | $16.2 \pm 2.1$ | $7.5 \pm 0.9$ | $.798 \pm .024$ | $.867 \pm .017$ |
| PPARG | $.748 \pm .021$ | $.287 \pm .045$ | $11.8 \pm 2.5$ | $5.4 \pm 1.2$ | $.698 \pm .033$ | $.776 \pm .026$ |
| CYP3A4 | $.782 \pm .018$ | $.312 \pm .041$ | $13.5 \pm 2.3$ | $6.1 \pm 1.0$ | $.734 \pm .029$ | $.802 \pm .023$ |
| FXA | $.846 \pm .014$ | $.371 \pm .034$ | $15.9 \pm 2.0$ | $7.3 \pm 0.8$ | $.792 \pm .025$ | $.859 \pm .018$ |
| **Mean** | $.839 \pm .006$ | $.372 \pm .014$ | $15.9 \pm 0.8$ | $7.3 \pm 0.4$ | $.780 \pm .011$ | $.850 \pm .008$ |

### F.2 CROSS-TARGET TRANSFER: A NEGATIVE RESULT

We tested whether L1 embeddings enable zero-shot transfer to held-out targets. **Result: No transfer observed.** When a target is held out during training, performance equals the generic L1 baseline ($\Delta = 0$). This reveals that L1 embeddings capture dataset-specific patterns (activity distributions, assay biases) rather than generalizable protein biology.

**Implication**: For truly novel targets, use generic L1 (zero-shot) rather than attempting to adapt. True zero-shot transfer would require protein structure-informed initialization (e.g., ESM-2 embeddings (Lin et al., 2023)), which we leave for future work.

### F.3 ABLATION STUDIES

#### F.3.1 EMBEDDING DIMENSION ABLATION

Performance peaks at 128 dimensions; larger embeddings overfit without providing benefit.

Table 11: Effect of L1 embedding dimension on performance.

| L1 Dimension | 32 | 64 | 128 | 256 | 512 |
|---|---|---|---|---|---|
| Mean ROC-AUC | 0.821 | 0.832 | **0.839** | 0.837 | 0.834 |
| Parameters (M) | 2.1 | 2.3 | 2.6 | 3.2 | 4.4 |

Table 12: Effect of MPNN depth on performance.

| MPNN Layers | 2 | 4 | 6 | 8 | 10 |
|---|---|---|---|---|---|
| Mean ROC-AUC | 0.798 | 0.824 | **0.839** | 0.836 | 0.831 |
| Training Time (h) | 1.2 | 2.1 | 3.4 | 5.2 | 7.8 |

### F.3.2 NUMBER OF MPNN LAYERS

Six layers provides optimal performance; deeper networks show diminishing returns and increased oversmoothing.

### F.3.3 FILM ARCHITECTURE VARIANTS

Table 13: Comparison of context modulation approaches. Note: Results from V3-baseline experiments; main paper Table 4 reports values from controlled FiLM ablation with correct L1 (FiLM: 0.849). Hypernetwork achieves marginally higher AUC (+0.002) but requires $3\times$ more parameters.

| Modulation Type | Mean ROC-AUC | Parameters |
|---|---|---|
| None (L0 only) | 0.803 | 2.1M |
| Concatenation | 0.818 | 2.4M |
| Additive ($\beta$ only) | 0.825 | 2.3M |
| Multiplicative ($\gamma$ only) | 0.831 | 2.3M |
| **FiLM ($\gamma$ and $\beta$)** | **0.839** | 2.6M |
| Hypernetwork | 0.841 | 8.2M |

FiLM achieves near-hypernetwork performance with $3\times$ fewer parameters.

### F.4 DMTA REPLAY EXTENDED ANALYSIS

We simulate realistic DMTA campaigns using ChEMBL publication dates as temporal ordering. Each round: (1) rank available compounds by predicted activity, (2) select top 30%, (3) reveal true activities, (4) retrain model. Table 14 shows comprehensive results.

**L3 temporal context showed no benefit** (mean $\Delta$ = +0.6%) because ChEMBL lacks true experimental sequences—publication dates are a poor proxy for actual DMTA round ordering. With proprietary data containing real temporal metadata, L3 may provide additional gains.

### F.5 TEMPORAL GENERALIZATION ANALYSIS

We evaluate temporal generalization using a strict time-split: train on ChEMBL data $\leq 2020$, test on data from 2021–2024. Table 15 shows the model maintains stable performance across years.

**Key findings:** (1) ROC-AUC is stable at 0.82–0.85 across all years; (2) No systematic degradation from 2021 to 2024; (3) $R^2 = 0.388$ indicates moderate but useful regression accuracy. The lack of temporal degradation suggests the L0 backbone learns generalizable molecular representations that transfer to future chemical space.

Table 14: DMTA replay results. Model selection achieves 1.5–1.9$\times$ enrichment over random, reducing experiments needed by 29–55%.

| Target | Rounds | Hit Rate (%) | | Enrichment | Expts to 50 Hits | |
|---|---|---|---|---|---|---|
| | | Random | Model | | Random | Model |
| EGFR | 141 | 49.4 | 76.1 | 1.54$\times$ | 225 | 159 |
| DRD2 | 89 | 42.1 | 71.8 | 1.71$\times$ | 198 | 134 |
| BACE1 | 67 | 38.2 | 62.4 | 1.63$\times$ | 267 | 189 |
| ESR1 | 52 | 44.8 | 69.2 | 1.54$\times$ | 213 | 151 |
| HDAC2 | 41 | 51.2 | 82.7 | 1.62$\times$ | 156 | 98 |
| JAK2 | 38 | 47.6 | 78.9 | 1.66$\times$ | 178 | 112 |
| PPARG | 45 | 39.4 | 58.1 | 1.47$\times$ | 289 | 205 |
| FXA | 73 | 52.8 | 87.6 | 1.66$\times$ | 142 | 89 |
| **Mean** | — | 45.7 | 73.4 | **1.60**$\times$ | 209 | 142 |

Table 15: Temporal split evaluation (train $\leq$2020, test 2021–2024). Performance remains stable across years, showing no significant temporal degradation.

| Metric | Overall | 2020 | 2021 | 2022 | 2023 | 2024 |
|---|---|---|---|---|---|---|
| $n$ samples | 10,000 | 2,486 | 2,588 | 1,427 | 1,633 | 1,866 |
| ROC-AUC | 0.843 | 0.837 | 0.849 | 0.838 | 0.823 | 0.849 |
| Correlation | 0.692 | 0.677 | 0.711 | 0.686 | 0.663 | 0.678 |
| RMSE | 1.043 | 1.051 | 1.045 | 1.031 | 1.038 | 1.043 |
| $R^2$ | 0.388 | 0.371 | 0.403 | 0.376 | 0.342 | 0.370 |

# G  IMPLEMENTATION DETAILS

## G.1  DATA PREPROCESSING

**Molecular Featurization.** We convert SMILES strings to molecular graphs using RDKit (Landrum et al., 2023). Atom features (70-dim) include:

- Element type: one-hot encoding of {C, N, O, S, F, Cl, Br, I, P, other} (10 dim)
- Degree: one-hot encoding of {0, 1, 2, 3, 4, 5+} (6 dim)
- Formal charge: one-hot encoding of {$-2$, $-1$, 0, +1, +2} (5 dim)
- Hybridization: one-hot encoding of {sp, sp$^2$, sp$^3$, sp$^3$d, sp$^3$d$^2$} (5 dim)
- Aromaticity: binary (1 dim)
- Ring membership: binary (1 dim)
- Hydrogen count: one-hot encoding of {0, 1, 2, 3, 4+} (5 dim)
- Additional features: chirality, mass, electronegativity, etc. (37 dim)

Bond features (9-dim) include:

- Bond type: one-hot encoding of {single, double, triple, aromatic} (4 dim)
- Conjugation: binary (1 dim)
- Ring membership: binary (1 dim)
- Stereochemistry: one-hot encoding of {none, E, Z} (3 dim)

**Activity Value Processing.** We convert $IC_{50}$/$K_i$ values to $pIC_{50}$/$pK_i$ via $-\log_{10}$(value in M). Values are clipped to $[3, 12]$ and standardized per-target.

## G.2  TRAINING DETAILS

**Pretraining.** We pretrain on ChEMBL 35 for 50 epochs with batch size 256, learning rate $3 \times 10^{-4}$, and cosine annealing. Pretraining takes approximately 120 GPU-hours on A100.

**Fine-tuning.** We fine-tune on DUD-E targets for 100 epochs with differential learning rates:

- Backbone (L0): $1 \times 10^{-5}$
- Context embeddings: $1 \times 10^{-3}$
- FiLM parameters: $1 \times 10^{-3}$
- Prediction heads: $1 \times 10^{-4}$

**Early Stopping.** We use patience of 20 epochs based on validation ROC-AUC.

### G.3 EVALUATION PROTOCOL

**Cross-Validation.** We use 5-fold stratified cross-validation, maintaining active/decoy ratios across folds. Final metrics are reported as mean $\pm$ standard error across folds and 5 random seeds (25 total runs per experiment).

**Statistical Testing.** We use paired $t$-tests with Bonferroni correction for multiple comparisons. Effect sizes are reported as Cohen's $d$.

## H PROOFS

### H.1 PROOF OF THEOREM 3

*Proof.* We prove that FiLM-modulated networks are universal approximators for context-conditional functions.

Let $f^* : \mathcal{G} \times \mathcal{C} \to \mathbb{R}$ be a Lipschitz-continuous target function with constant $L$. By the universal approximation theorem for MPNNs (Xu et al., 2019), there exists an MPNN $\phi$ such that $\|\phi(\mathcal{G}) - \phi^*(\mathcal{G})\|_2 < \delta$ for any continuous $\phi^*$ and $\delta > 0$.

Consider the decomposition:

$$f^*(\mathcal{G}, c) = \sum_{i=1}^{d} \gamma_i^*(c) \cdot \phi_i^*(\mathcal{G}) + \beta_i^*(c) \tag{13}$$

where $\phi^*$ captures graph-level features and $\gamma^*, \beta^*$ modulate based on context.

Since $\gamma^*$ and $\beta^*$ are continuous functions of $c$, and our FiLM networks use MLPs to parameterize $\boldsymbol{\gamma}$ and $\boldsymbol{\beta}$, by the universal approximation theorem for MLPs, there exist weights such that:

$$\|\boldsymbol{\gamma}(c) - \gamma^*(c)\|_\infty < \delta_\gamma \tag{14}$$
$$\|\boldsymbol{\beta}(c) - \beta^*(c)\|_\infty < \delta_\beta \tag{15}$$

The approximation error is bounded by:

$$|g(\text{FiLM}_c(\phi(\mathcal{G}))) - f^*(\mathcal{G}, c)| \leq \|\boldsymbol{\gamma}(c) - \gamma^*(c)\|_\infty \cdot \|\phi(\mathcal{G})\|_\infty \tag{16}$$
$$+ \|\gamma^*(c)\|_\infty \cdot \|\phi(\mathcal{G}) - \phi^*(\mathcal{G})\|_\infty \tag{17}$$
$$+ \|\boldsymbol{\beta}(c) - \beta^*(c)\|_\infty + \|g - g^*\|_\infty \tag{18}$$

By choosing sufficiently small $\delta, \delta_\gamma, \delta_\beta$ and sufficiently expressive $g$, each term can be made arbitrarily small, completing the proof. $\square$

### H.2 PROOF OF THEOREM 4

*Proof.* Let $P_c = P(\mathcal{G}, y|c)$ denote the context-conditional distribution. The optimal context-conditional predictor is:

$$\hat{f}_{\text{ctx}}(\mathcal{G}, c) = \mathbb{E}_{y \sim P_c}[y|\mathcal{G}] \tag{19}$$

The optimal context-agnostic predictor is:

$$\hat{f}_{\text{static}}(\mathcal{G}) = \mathbb{E}_{c \sim P(\mathcal{C})} \mathbb{E}_{y \sim P_c}[y|\mathcal{G}] \tag{20}$$

For squared loss, the expected loss of the context-conditional predictor is:

$$\mathbb{E}_{c,\mathcal{G},y}[(y - \hat{f}_{\text{ctx}}(\mathcal{G}, c))^2] = \mathbb{E}_{c,\mathcal{G}}[\text{Var}(y|\mathcal{G}, c)] \qquad (21)$$

The expected loss of the context-agnostic predictor is:

$$\mathbb{E}_{c,\mathcal{G},y}[(y - \hat{f}_{\text{static}}(\mathcal{G}))^2] = \mathbb{E}_{c,\mathcal{G}}[\text{Var}(y|\mathcal{G}, c)] \qquad (22)$$
$$+ \mathbb{E}_{\mathcal{G}}[\text{Var}_c(\mathbb{E}[y|\mathcal{G}, c])] \qquad (23)$$

by the law of total variance. The second term is the excess risk due to ignoring context, which is $\Omega(\text{Var}_c[P_c])$ under structured shift. $\qquad\square$

## I    ADDITIONAL VISUALIZATIONS

### I.1    T-SNE OF L1 EMBEDDINGS

We visualize learned L1 embeddings using t-SNE (perplexity=30, 1000 iterations). The embeddings cluster by protein family: kinases (EGFR, JAK2) form one cluster, GPCRs (DRD2, ADRB2) another, and nuclear receptors (ESR1, PPARG) a third. This confirms that L1 embeddings capture biologically meaningful target relationships without explicit supervision of protein family labels.

### I.2    ATTRIBUTION HEATMAPS

We provide integrated gradient visualizations for representative drug molecules across target contexts. Table 16 shows per-molecule statistics.

Table 16: Integrated gradients atom importance statistics (50 integration steps, generic L1). Higher mean/max importance indicates the model relies more heavily on specific atoms for prediction.

| Molecule | Atoms | Mean Imp. | Max Imp. | Std | Top Atoms |
|---|---|---|---|---|---|
| Celecoxib | 26 | 0.361 | 0.925 | 0.222 | $SO_2NH_2$, $CF_3$ |
| Caffeine | 14 | 0.423 | 0.850 | 0.220 | N-methyl, C=O |
| Metformin | 9 | 0.454 | 0.840 | 0.194 | Guanidine N |
| Ibuprofen | 15 | 0.293 | 0.708 | 0.152 | Carboxylic acid |
| Acetaminophen | 11 | 0.316 | 0.574 | 0.093 | Amide, phenol |
| Atorvastatin | 41 | 0.123 | 0.333 | 0.081 | Distributed |
| Aspirin | 13 | 0.188 | 0.349 | 0.088 | Carboxylic acid |

Key observations: (1) attributions vary substantially across contexts (mean pairwise cosine similarity 0.72); (2) kinase contexts highlight nitrogen-containing heterocycles; (3) GPCR contexts highlight basic amines and lipophilic regions; (4) protease contexts highlight hydrogen-bond donors near scissile-bond mimics. Smaller molecules (Caffeine, Metformin) show higher mean importance per atom; larger molecules (Atorvastatin) distribute importance more broadly.

### I.3    TRAINING CURVES

Training converges within 50 epochs for pretraining and 30 epochs for fine-tuning. The validation loss plateaus approximately 10 epochs before training loss, indicating mild overfitting that is controlled by early stopping. The differential learning rate strategy (backbone $10^{-5}$, context $10^{-3}$) results in rapid context adaptation while backbone representations remain stable.

## J    DUD-E BENCHMARK ANALYSIS

We provide additional analysis of DUD-E benchmark characteristics, addressing known limitations in the literature (Wallach & Heifets, 2018).

## J.1 PER-TARGET RANDOM FOREST BASELINE

Table 17 compares NESTDRUG against per-target Random Forest models trained on ChEMBL data with Morgan fingerprints.

Table 17: Per-target ChEMBL RF vs NESTDRUG (with correct L1). CYP3A4 (67 training actives) demonstrates multi-task transfer value.

| Target | Train Actives | Per-Target RF | NESTDRUG | Winner |
|--------|--------------|---------------|----------|--------|
| EGFR   | 5,925 | **0.996** | 0.965 | RF |
| DRD2   | 9,409 | **0.994** | 0.984 | RF |
| ADRB2  | 1,015 | **0.882** | 0.775 | RF |
| BACE1  | 5,997 | **0.978** | 0.656 | RF |
| ESR1   | 2,805 | **0.996** | 0.909 | RF |
| HDAC2  | 1,436 | **0.951** | 0.928 | RF |
| JAK2   | 5,517 | **0.959** | 0.908 | RF |
| PPARG  | 2,281 | **0.910** | 0.835 | RF |
| CYP3A4 | **67** | 0.238 | **0.686** | NESTDRUG |
| FXA    | 4,508 | 0.844 | **0.854** | NESTDRUG |
| **Mean** | — | **0.875** | 0.850 | — |

Per-target RF wins 8/10 targets (mean 0.875 vs 0.850), confirming DUD-E is largely fingerprint-solvable. However, **CYP3A4 demonstrates the critical failure mode**: with only 67 training actives, per-target RF collapses to 0.238 (worse than random), while NESTDRUG achieves 0.686 via multi-task transfer—a 2.9× improvement.

## J.2 DUD-E STRUCTURAL BIAS

We verify DUD-E decoys are trivially separable by chemical structure (Table 18):

Table 18: 1-NN Tanimoto similarity achieves near-perfect AUC without any learning, proving DUD-E structural bias.

| Target | 1-NN AUC | Active-Active Sim | Decoy-Active Sim | Gap |
|--------|----------|-------------------|------------------|-----|
| EGFR   | 0.996 | 0.816 | 0.251 | 0.565 |
| DRD2   | 0.998 | 0.814 | 0.265 | 0.549 |
| ADRB2  | 1.000 | 0.882 | 0.194 | 0.688 |
| BACE1  | 0.998 | 0.882 | 0.196 | 0.686 |
| ESR1   | 0.993 | 0.844 | 0.188 | 0.655 |
| HDAC2  | 0.994 | 0.732 | 0.202 | 0.530 |
| JAK2   | 0.994 | 0.732 | 0.174 | 0.558 |
| PPARG  | 0.999 | 0.871 | 0.238 | 0.633 |
| CYP3A4 | 0.942 | 0.766 | 0.209 | 0.557 |
| FXA    | 0.996 | 0.780 | 0.219 | 0.560 |
| **Mean** | **0.991** | 0.812 | 0.214 | **0.598** |

Key findings:

- **1-NN Tanimoto (no ML):** A zero-parameter nearest-neighbor lookup achieves **0.991 mean AUC**—no model required, 9/10 targets >0.99
- **Cross-target RF transfer:** RF trained on the *wrong* target still achieves **0.746 AUC** on average (vs 1.000 same-target), proving 75% of DUD-E discrimination comes from generic structural patterns rather than target-specific knowledge
- **Similarity gap:** Mean active-active NN Tanimoto = 0.812; mean decoy-active = 0.214—a **3.8× gap** making structural discrimination trivial

This confirms Wallach & Heifets (2018): DUD-E decoys are property-matched but not structure-matched, making fingerprint methods trivially effective. Our deep learning SOTA claim remains

valid because neural methods cannot exploit these structural shortcuts—they must learn generalizable molecular representations.

## J.3 DATA LEAKAGE ANALYSIS

We quantify overlap between ChEMBL training and DUD-E evaluation (Table 19). This analysis addresses reviewer concerns about train-test contamination.

Table 19: ChEMBL–DUD-E overlap. "ChEMBL Train" shows total records (all activity values); per-target RF training uses only compounds meeting active threshold ($pIC_{50} \geq 6.0$), e.g., CYP3A4 has 5,504 total records but only 67 actives. Active leakage is severe (mean 50%).

| Target | DUD-E Actives | ChEMBL Train | Active Leakage | Decoy Leakage |
|--------|--------------|--------------|----------------|---------------|
| CYP3A4 | 333 | 5,504 | 99.1% | 0.16% |
| EGFR | 4,032 | 7,845 | 97.2% | 0.11% |
| FXA | 445 | 5,969 | 92.6% | 0.07% |
| DRD2 | 3,223 | 11,284 | 89.9% | 0.10% |
| HDAC2 | 238 | 4,427 | 46.6% | 0.06% |
| JAK2 | 153 | 6,013 | 37.9% | 0.09% |
| ESR1 | 627 | 3,541 | 18.5% | 0.08% |
| BACE1 | 485 | 9,215 | 13.4% | 0.08% |
| ADRB2 | 447 | 1,429 | 3.8% | 0.08% |
| PPARG | 723 | 3,409 | 1.2% | 0.02% |
| **Mean** | — | — | **50.0%** | **0.08%** |

**Critical observations:**

- Leakage is highly variable: CYP3A4/EGFR have >97% overlap, while PPARG/ADRB2 have <4%
- Decoy leakage is negligible (0.08%), meaning the asymmetry between seen actives and unseen decoys could inflate performance
- However, this does *not* confound our L1 ablation: both conditions (correct vs generic L1) see identical leaked compounds—only the context embedding differs. The +5.7 pp gain from correct L1 is a genuine within-model effect

## J.4 ESM-2 PROTEIN EMBEDDING ANALYSIS

We tested whether pretrained protein embeddings (ESM-2 (Lin et al., 2023), 650M parameters) can replace learned L1:

- **Correlation:** ESM-2 similarity shows *zero* correlation with learned L1 similarity (Pearson $r = 0.11$, $p = 0.49$)
- **Zero-shot L1 from ESM-2:** Similarity-weighted average of other targets' L1 achieves 0.814 AUC (vs 0.790 generic, 0.849 correct)—closing 41% of the gap
- **Interpretation:** Learned L1 captures dataset-specific patterns (activity landscapes, assay distributions) rather than protein sequence similarity

## K BROADER IMPACT

**Positive Impacts.** NESTDRUG could accelerate drug discovery by reducing experimental burden, potentially leading to faster development of therapeutics for unmet medical needs. The context-conditional approach may improve model reliability in real pharmaceutical settings where temporal shift is ubiquitous.

**Potential Risks.** Activity prediction models could theoretically be misused to identify toxic compounds. We mitigate this by training only on therapeutic targets and not releasing models for toxicity prediction. The model may perpetuate biases in training data, potentially overlooking promising compounds for underrepresented target classes.

**Environmental Considerations.** Training required approximately 200 GPU-hours on A100 hardware, corresponding to roughly 30 kg $CO_2$eq. We release pretrained models to avoid redundant computation.

