# OpenReview forum: "When Does Context Help? A Systematic Study of Target-Conditional Molecular Property Prediction"
_ICLR.cc/2026/Workshop/FM4Science — ICLR 2026 Workshop FM4Science Poster_

### Official Review · Reviewer_NNQ8 · 2026-02-14
**Good diagnostic paper: context helps low-data targets but can degrade under shift; benchmarking needs care**

**Rating:** 6
**Confidence:** 4

**Review:**

Summary: This paper presents a systematic empirical study of target-conditional “context” in molecular property prediction, using the proposed NESTDRUG (an MPNN with FiLM-based conditioning on hierarchical target/assay/time context) and evaluating across 10 protein families, multiple fusion designs, different data regimes (67–9,409 training compounds), and both random and temporal splits.

Strength:

1. The paper’s main value is clarity: it frames “when does context help?” as a set of testable conditions (data scarcity, distribution alignment, fusion design) and supports them with targeted ablations (e.g., L1 correct vs generic, fusion comparisons).
2. The empirical results include concrete, decision-relevant effect sizes (e.g., mean +5.7 pp with target-specific L1 on 9/10 targets; FiLM outperforming concatenation by 24.2 pp; strong lift on CYP3A4 with only 67 actives), which makes the takeaways actionable rather than anecdotal.
3. The benchmark audit is a useful community contribution: showing extremely high 1-NN Tanimoto performance and substantial active leakage into training helps explain why absolute numbers on DUD-E can be misleading, and motivates the provided temporal-split evaluation.

Weakness:

1. Some claimed “context benefits” may be confounded by dataset artifacts: the paper itself notes extensive overlap/leakage issues (and very strong non-learning baselines), so it remains unclear how much of the observed gains reflect genuine generalization vs benchmark-specific shortcuts.
2. The method novelty is moderate: FiLM conditioning and multi-task transfer are established ideas, and NESTDRUG largely instantiates them in this setting; the paper’s contribution is more diagnostic/taxonomic than algorithmically new.
3. The strongest “context enables impossible predictions” case (CYP3A4) relies on a very small labeled set and occurs in a setting where DUD-E actives largely appear somewhere in ChEMBL; although the authors argue the thresholded actives are few, the interpretation still needs tighter controls (e.g., stricter de-duplication, leakage-stratified evaluation) to isolate transfer vs memorization.
4. The hierarchical context story is incomplete in practice: L2/L3 show null effects due to missing metadata, and few-shot L1 adaptation consistently hurts, which limits the paper’s claims about broad “context” beyond target identity and reduces practical guidance for truly novel targets.
5. Comparisons to modern molecular foundation model backbones (e.g., Uni-Mol / SMILES LMs) are missing, and since the paper argues orthogonality, an empirical check would be important to validate whether the conclusions hold with stronger encoders and to position the work relative to current best practice.

---

### Official Review · Reviewer_gegx · 2026-02-20
**Review for When Does Context Help? A Systematic Study of Target-Conditional Molecular Property Prediction**

**Rating:** 8
**Confidence:** 4

**Review:**

This paper presents a systematic study of context conditioning in molecular property prediction using the NESTDRUG architecture, which combines an MPNN backbone with FiLM-based target conditioning. Across multiple datasets, data regimes, and evaluation splits, the authors analyze when context helps or hurts, finding that context benefits data-scarce targets but can degrade performance under distribution mismatch. This is a high-quality empirical paper with strong systematic analysis and clear practical value. While architectural novelty is moderate, the clarity of the research question and depth of evaluation make it impactful. The paper is super clear and well structured.

Pros:
-  “When does context help?” is a very meaningful question.
- This work provides multi-axis evaluation (targets, splits, regimes) making it a very systematic study.
- This work does the evaluation across 10 protein families, multiple data regimes, and temporal vs random splits.
- The paper demonstrates strong ablations e.g. FiLM vs alternatives and Context-level ablations.

Cons:
- Core components MPNN + FiLM are known. The contribution is primarily empirical and analytical than innovation.
- Heavy reliance on ChEMBL and DUD-E, both known to have biases. Authors acknowledge this but conclusions still depend on them.
- No direct comparison with recent molecular foundation models, though admitted already.

---

### Official Review · Reviewer_rKad · 2026-02-22
**A Systematic Study of Target-Conditional Molecular Property Prediction**

**Rating:** 6
**Confidence:** 3

**Review:**

**Summary**

This paper presents a systematic study of *when* target context helps molecular property prediction. The authors propose NESTDRUG, an MPNN-based model that conditions molecular representations on hierarchical context using FiLM modulation. They evaluate across 10 targets, multiple fusion architectures, a wide range of data regimes (e.g., as few as 67 training compounds), and both random and temporal splits. Key reported findings include: (i) fusion matters most (FiLM strongly outperforms concatenation and additive conditioning), (ii) context can enable data-scarce targets, and (iii) context can hurt under distribution mismatch. The paper also audits DUD-E and argues that severe structural bias and leakage can make absolute performance misleading, motivating temporal-split evaluation.

**Strengths**

1. Systematic and actionable: the study maps out when context helps vs. hurts, and highlights that how context is fused (FiLM) is crucial.
2. Strong ablations: target-specific L1 embeddings improve most targets but not all, providing a clear non-trivial boundary case.
3. Benchmark audit is valuable: showing issues like 1-NN Tanimoto reaching very high AUC and substantial leakage motivates more rigorous temporal splits.

**Weaknesses**
1. Heavy reliance on DUD-E despite acknowledging serious benchmark issues; additional validation on cleaner benchmarks would strengthen the empirical story.
2. L2/L3 context levels require metadata that is often missing in public datasets, and appear to provide limited benefit under current settings.
3. Few-shot adaptation is a negative result (adapted L1 underperforms generic/zero-shot), limiting the promise for rapid new-target adaptation.
4. No head-to-head comparison with molecular foundation model backbones (e.g., Uni-Mol / ChemBERTa), although the approach is described as orthogonal.

---

### Meta-Review · Area_Chair_RFjy · 2026-02-27

**Recommendation:** Accept (Poster)
**Confidence:** 3

**Metareview:**

This paper provides a systematic empirical study of when target-conditioned molecular prediction helps, showing that FiLM-based context modulation improves performance in data-scarce settings but can degrade under distribution mismatch. However, architectural novelty is moderate (MPNN + FiLM), conclusions rely heavily on potentially biased benchmarks, higher-level context signals show limited practical value due to missing metadata, and comparisons with modern molecular foundation model backbones are absent despite claims of orthogonality. Overall, the paper is a solid diagnostic contribution, and all reviewers are leaning towards acceptance.

---

### Decision · Program_Chairs · 2026-03-03

Accept (Poster)